# Biodegradable Covalently Crosslinked Poly[*N*-(2-Hydroxypropyl) Methacrylamide] Nanogels: Preparation and Physicochemical Properties

**DOI:** 10.3390/polym16020263

**Published:** 2024-01-17

**Authors:** Jana Kousalová, Petr Šálek, Ewa Pavlova, Rafał Konefał, Libor Kobera, Jiří Brus, Olga Kočková, Tomáš Etrych

**Affiliations:** Institute of Macromolecular Chemistry, Academy of Sciences of the Czech Republic, Heyrovského Nám. 2, 162 00 Prague, Czech Republic; kousalova@imc.cas.cz (J.K.); pavlova@imc.cas.cz (E.P.); konefal@imc.cas.cz (R.K.); kobera@imc.cas.cz (L.K.); brus@imc.cas.cz (J.B.); kockova@imc.cas.cz (O.K.); etrych@imc.cas.cz (T.E.)

**Keywords:** biodegradable, dispersion polymerization, glutathione, *N*-(2-hydroxypropyl) methacrylamide, nanogel

## Abstract

Recently, suitably sized polymer-based nanogels containing functional groups for the binding of biologically active substances and ultimately degradable to products that can be removed by glomerular filtration have become extensively studied systems in the field of drug delivery. Herein, we designed and tailored the synthesis of hydrophilic and biodegradable poly[*N*-(2-hydroxypropyl) methacrylamide-co-*N*,*N*′-bis(acryloyl) cystamine-*co-*6-methacrylamidohexanoyl hydrazine] (PHPMA-BAC-BMH) nanogels. The facile and versatile dispersion polymerization enabled the preparation of nanogels with a diameter below 50 nm, which is the key parameter for efficient and selective passive tumor targeting. The effects of the *N*,*N*′-bis(acryloyl) cystamine crosslinker, polymerization composition, and medium including H_2_O/MetCel and H_2_O/EtCel on the particle size, particle size distribution, morphology, and polymerization kinetics and copolymer composition were investigated in detail. We demonstrated the formation of a 38 nm colloidally stable PHPMA-BAC-BMH nanogel with a core–shell structure that can be rapidly degraded in the presence of 10 mM glutathione solution under physiologic conditions. The nanogels were stable in an aqueous solution modeling the bloodstream; thus, these nanogels have the potential to become highly important carriers in the drug delivery of various molecules.

## 1. Introduction

Nanogels represent three-dimensional nanoscale structures with physically and/or covalently crosslinked polymer networks [1]. These unique nanosystems are characterized by their hydrophilicity, an ability to swell in aqueous media, softness, porous structure, biocompatibility, stimuli-responsive behavior such as a change in pH, temperature, or ionic strength, as well as biodegradability. In particular, biodegradable nanogels as delivery systems have attracted much attention in the biotechnological and biomedical fields due to their desired ability to degrade in intracellular environments or to react with various external stimuli [2]. There are a variety of constituents serving as building blocks for final polymer nanogels, with N-isopropylacrylamide being one of the most common monomers for the preparation of temperature-responsive nanogel delivery systems due to its tunable lower critical solution temperature [3,4]. Also, other acrylic- and acrylamide-based monomers are used for the fabrication of functional nanogels including acrylic acid, methacrylic acid, 2-hydroxyethyl methacrylate, acrylamide, and *N*,*N*-dimethylacrylamide [2,5,6,7]. A variety of methods have been developed such as inverse mini- or microemulsion, precipitation, dispersion, controlled radical polymerizations, nanoprecipitation, gelation, and microfluidic methods. However, at present, the inconsistencies between batches obtained by some previously mentioned techniques and the limited manufacturing efficiency represent the most significant obstacles to be overcome for the extensive utilization and integration of nanogels in preclinical research and clinics [2,8].

Dispersion polymerization is a promising method for the preparation of nanogels because it represents a simple and single batch process yielding particles with an average diameter from 0.1 to 15 µm. The polymerization is performed in the presence of monomer(s), suitable stabilizer(s), an initiator, and a thermodynamically good solvent for all reactants but not the resulting polymer. Dispersion polymerization in organic media producing nonaqueous dispersions was first introduced in 1975, followed by dispersion polymerization in polar solvents and the study of various parameters to produce polymeric particles with controlled size and shape [9]. Another important progress made in dispersion polymerization was the preparation of polymer particles in aqueous alcohol, resulting in smaller polymer particles with increased molecular weight, as well as hydrophilic polymer particles, the most important being thermoresponsive poly(*N*-isopropylacrylamide) particles [10,11]. Uyama et al. prepared hydrophilic poly(*N*-vinylformamide) particles in polar media using poly(2-ethyl-2-oxazoline) as a stabilizer [12]. In 1998, Takahashi et al. reported the dispersion polymerization of glycidyl methacrylate in a solvent mixture of methanol and *N*,*N*-dimethylformamide stabilized with poly(*N*-vinylpyrrolidone) [13]. Horák studied various reaction parameters of dispersion polymerization to produce micron-sized poly(2-hydroxyethyl methacrylate) particles [14].

Poly[*N*-(2-hydroxypropyl) methacrylamide] (PHPMA) is a hydrophilic and biocompatible polymer typically used in biomedical and pharmaceutical applications, such as drug delivery, targeted therapy, imaging, and tissue engineering, due to its unique and tunable properties [15]. PHPMA has been thoroughly studied as a suitable non-fouling polymer for the synthesis of polymeric nanosystems, e.g., micelles and polymersomes, forming their outer hydrophilic shell. PHPMA as the outer shell renders polymeric micelles hydrophilic and non-cytotoxic with improved colloidal stability, reactive functionality, reduced undesired interaction with proteins and cells, and prolonged circulation time [16]. For example, Tang et al. designed an HPMA copolymer conjugated with doxorubicin self-assembled into stimuli-responsive, biodegradable 15 and 20 nm nanoparticles with long-lasting blood circulation and the ability to inhibit tumor growth [17]. Phan et al. reported aqueous RAFT polymerization-induced self-assembly of a PHPMA-*b*-poly[*N*-(2-methylthio)ethyl acrylamide] block copolymer into differently shaped polymersomes depending on the length of the hydrophilic or hydrophobic block [18]. Our group prepared PHPMA-based nanogels by dispersion polymerization and their fluorescently labeled analogs possessed non-cytotoxic properties and were well-distributed in the cytosol where they colocalized with lysosomes [19]. However, the application of these polymer nanosystems is limited by their persistent nature; thus, they cannot be removed after fulfilling their carrier role. This may be overcome by the use of biodegradable nanoparticles. For example, *N*,*N*′-bis(acryloyl) cystamine (BAC) containing disulfide bonds is used as a crosslinker that can undergo cleavage under physiologic reducing conditions. Typically, glutathione (GSH) is used as a suitable reducing agent with an extracellular concentration of 2–20 µM and intracellular concentration of 0.5–10 mM. Moreover, glutathione activity is strongly connected to the redox proteins thioredoxin 1 and 2 to maintain the cellular redox balance. Wutzel et al. successfully synthesized ~150 nm reduction-sensitive PHPMA nanoparticles crosslinked with BAC by RAFT polymerization in inverse emulsion [20]. The optimal nanoparticle size for the highest cellular uptake is around 50 nm or even lower [21] for drug delivery systems and cellular uptake.

In this study, we report the facile preparation of ~40 nm biodegradable poly[*N*-(2-hydroxypropyl) methacrylamide-co-*N*,*N*′-bis(acryloyl) cystamine-*co-*6-methacrylamidohexanoyl hydrazine] (PHPMA-BAC-BMH) nanogels by dispersion polymerization in aqueous media. For this purpose, we studied the effects of BAC concentration (5 to 25 wt%), different ratios of H_2_O/2-methoxyethanol (MetCel) and H_2_O/2-ethoxyethanol (EtCel) on the size, particle size distribution, morphology, and composition of the final nanogels. The biodegradability of the PHPMA-BAC-BMH nanogel was evaluated in the presence of 10 mM GSH solution under physiologic conditions by TEM analysis, showing that the PHPMA-BAC-BMH nanogel was rapidly degraded within 24 h.

## 2. Materials and Methods

### 2.1. Materials

*N*,*N*′-bis(acryloyl) cystamine (BAC), glutathione (GSH), 2-methoxyethanol (MetCel), phosphate-buffered saline tablets, potassium persulfate (KPS; crystallized from water), poly(vinylpyrrolidone) K90 (PVP), methacryloyl chloride, 1-aminopropan-2-ol, 6-aminohexanoic acid, *tert*-butyl carbazatem, methyl-6-aminohexanoate, and hydrazine hydrate were purchased from Sigma Aldrich (St. Louis, MO, USA). Poly(vinyl alcohol) 25/140 (PVA) was obtained from Wacker Chemie (München, Germany). 2-Ethoxyethanol (EtCel) and CH_3_OH were purchased from VWR (Radnor, PA, USA). Dichloromethane, NaOH, and Na_2_CO_3_ were purchased from Lachema (Brno, Czech Republic). Tetrahydrofurane was purchased from Lachner (Neratovice, Czech Republic). EDC.HCl was purchased from Gen Script (Piscataway, NJ, USA). For the A4F characterization, the water of MilliQ purity was prepared by laboratory water purification system Milli-Q^®^ IQ 7000 (Merck KGaA, Darmstadt, Germany), and sodium azide (NaN_3_; ≥99%, Lach-Ner, Neratovice, Czech Republic) was used as an antibacterial agent.

### 2.2. Synthesis of Monomers

*N*-(2-hydroxypropyl) methacrylamide (HPMA) was synthesized by the reaction of methacryloyl chloride with 1-aminopropan-2-ol in the presence of NaOH in dichloromethane as described previously [22]. Elemental analysis: calc./found: C 57.7%/57.96%, H 8.33%/8.64%, N 8.33%/8.44%. ^1^H NMR (300 MHz, DMSO-d6): δ 7.79 (s, 1H, NH); δ 5.65 (s, 1H, CH_2_=); δ 5.30 (s, 1H, CH_2_=); δ 4.69 (s, 1H, OH); δ 3.74–3.65 (m, 1H, CHOH); δ 3.1–3.0 (m, 2H, CH_2_N); δ 1.85 (s, 1H, CH_3_C); δ 1.01 (d, 3H, CH_3_CH).

*N*-(*tert*-butoxycarbonyl)-*N*′-(6-methacrylamidohexanoyl) hydrazine (BocBMH) was synthesized in two steps, as described previously [23]. In the first step, *N*-methacryloyl-6-aminohexanoic acid was prepared by the reaction of methacryloyl chloride with 6-aminohexanoic acid in the presence of NaOH in dichloromethane; then, BocBMH was prepared by reaction of 6-methacrylamidohexanoic acid with tert-butyl carbazate in the presence of EDC·HCl in tetrahydrofurane. Elemental analysis: calc./ found C 57.7%/58.66%, H 8.33%/8.44%, N 13.46%/13.16%. ^1^H NMR (300 MHz, DMSO-d6): δ 9.42 (s, 1H, NHCOO); δ 8.62 (s, 1H, NHCO); δ 7.84 (s, 1H, NHCO); δ 5.61 (s, 1H, CH_2_=); δ 5.27 (s, 1H, CH_2_=); δ 3.08–3.04 (q, 2H, CH_2_NH); δ 2.1 (t, 2H, CH_2_CO); δ 1.83 (s, 3H, CH_3_C); δ 1.6–1.37 (m, 13H, CH_2_CH_2_, (CH_3_)_3_C); δ 1.30–1.20 (m, 2H, CH_2_).

BMH was synthesized in two steps, as described previously [24]. In the first step, methyl-N-methacryloyl-6-aminohexanoate was prepared by the reaction of methacryloyl chloride with methyl-6-aminohexanoate in the presence of Na2CO3 in dichloromethane; then, BMH was prepared by reaction of methyl 6-methacrylamidohexanoate with hydrazine hydrate in CH_3_OH. Elemental analysis: calc./found C 56.08%/56.32%, H 9.06%/8.98%, N 19.83%/19.70%. 1H NMR (400 MHz, DMSO-d6): δ 8.89 (s, 1H, NHCO); δ 7.85 (s, 1H, NHCO); δ 5.60 (s, 1H, CH_2_=); δ 5.28 (s, 1H, CH_2_=); δ 4.11 (s, 2H, NH_2_NH); δ 3.05 (t, 2H, CH_2_CO); δ 1.99 (t, 2H, CH_2_NH); δ 1.83 (s, 3H, CH_3_C); δ 1.49–1.37 (m, 4H, CH_2_CH_2_); δ 1.24–1.22 (m, 2H, CH_2_).

### 2.3. Dispersion Polymerization of N-(2-Hydroxypropyl) Methacrylamide, N-(Tert-butoxycarbonyl)-N′-(6-methacrylamidohexanoyl) Hydrazine, and N,N′-Bis(acryloyl) Cystamine

Poly[*N*-(2-hydroxypropyl) methacrylamide-*co*-*N*-(tert-butoxycarbonyl)-*N*′-(6-methacrylamidohexanoyl) hydrazine-*co*-*N*,*N*′-bis(acryloyl) cystamine] nanogels (NG1-10) and poly[*N*-(2-hydroxypropyl) methacrylamide-*co*-6-methacrylamidohexanoyl hydrazine-*co*-*N*,*N*′-bis(acryloyl) cystamine] (NG11) were prepared according to the published procedure with a slight modification and Figure 1 [19]. Briefly, dispersion polymerization was performed in a glass 100 mL reaction vessel equipped with an anchor-type stirrer. PVP (0.2 g) and PVA (0.2 g) were added to MetCel (10 g) and water (60 g) (Solution 1) in the reaction vessel and stirred at 80 °C for 1 h. Meanwhile, HPMA (1.386 g), BAC (0.0815 g), and BocBMH (0.163 g) were added to MetCel (6 g) and water (4 g) (Solution 2) and stirred at RT for 1 h. Then, Solution 2 and KPS (0.16 g) were added to Solution 1 in the reaction vessel. The polymerization was allowed to proceed at 80 °C for 20 h while stirring (400 rpm). At the end of the reaction, the polymerization mixture was concentrated on a rotary evaporator, and then transferred to a dialysis membrane (cut-off < 100 kDa) for dialysis against water for 7 days to remove residual impurities (stabilizers, MetCel, linear polymer, unreacted monomers). Finally, the PHPMA-(Boc)BMH-BAC nanogel was freeze-dried.

### 2.4. Study of Nanogel Degradation

PHPMA-BMH-BAC nanogel (NG11; 3 mg·mL^−1^) was dispersed in PBS buffer (pH 7.4) in the presence of GSH (10 mmol·L^−1^) and incubated at 37 °C for 24 h. The process of degradation was monitored by TEM analysis at different time intervals (0, 1, 3, 5, and 24 h) to visualize NG11 degradation. The samples were negatively stained with uranyl acetate (2 wt.%) before the observation. The degradation study was performed in duplicate.

### 2.5. Characterization

The morphology, size, and particle size distribution of nanogels were studied using a Tecnai G2 Spirit Twin 12 transmission electron microscope (TEM; FEI; Brno, Czech Republic). The samples (3 mg·mL^−1^) were negatively stained with uranyl acetate (2 wt.%) and the number-average diameter (*D_n_*), weight-average diameter (*D*_w_), and dispersity (*Ð* = *D*_w_/*D_n_*) were calculated using ImageJ 1.54h software by counting at least 300 hydrogel nanoparticles as follows:(1)$$D_{n}=\frac{\sum n_{i}D_{i}}{\sum n_{i}}$$
(2)$$D_{w}=\frac{\sum n_{i}D_{i}^{4}}{\sum n_{i}D_{i}^{3}}$$
where *n_i_* and *D*_i_ are the number and diameter of the *i*-th microsphere, respectively.

Dynamic light scattering (DLS) measurements were performed at 25 °C using a ZEN 3600 Zetasizer Nano Instrument (Malvern, Instruments; Malvern, UK) at 633 nm and 173° detection angle. Intensity size and size distribution were obtained from the correlation function using CONTIN analysis available in the Malvern Zetasizer Software 7.11. The hydrodynamic diameters (*D*_h_) of all nanogels were calculated using the Stokes–Einstein equation. The concentration of samples was 1 mg.mL^−1^ in Q-H_2_O.

Asymmetric flow field flow fractionation (A4F) was used to determine the diameter of gyration (*D*_g_) of the samples using the refractive index increment (d*n*/d*c*) for PHPMA-based polymers d*n*/d*c* = 0.17 [19]. The solvent and sample delivery part of the system consisted of an Agilent G1310A pump, a G1322A degasser, and a G1329A autosampler. The field-flow fractionation long channel was assembled with a 350 μm spacer and a regenerated cellulose membrane with a cutoff of 10,000 g/mol. Three detectors were used in series: a Spectromonitor 3200 UV/VIS unit (Thermo Separation Products, Fremont, CA, USA), a Wyatt Optilab-rEX RI detector, and a Wyatt Dawn 8+ multiangle light-scattering unit. Wyatt software Astra V (version 5.3.4.15) controlled all system components through a Wyatt ECLIPSE 3+ unit. Water with sodium azide (0.2 g·L^−1^) was used as a solvent. The samples were filtered with a 0.45 μm PVDF filter before injection. A4F measurements were conducted with a constant detector flow rate of 0.6 mL/min. The focusing time was 5 min at a cross-flow of 1.5 mL/min. The injection flow was 0.2 mL/min, and 100 µL of the sample was injected in all cases. After the focusing step, the cross-flow was linearly decreased from 1.0 mL/min to 0.1 mL/min in 60 min and was then kept constant at 0.1 mL/min for the next 15 min, followed by 15 min without cross-flow.

^1^H NMR kinetics spectra were acquired with a Bruker Avance III 600 spectrometer operating at 600.2 MHz using “zgdelay” pulse program [25] at 353K with D_2_O/MetCel (1 mL) as a solvent under the same condition as for the preparation of NG11 by dispersion polymerization. The width of the 90° pulse was 18 μs, the relaxation delay was 10 s, and the acquisition time was 2.18 s in 8 scans. Kinetics measurements were provided for 20 h with time points every 0.5 h. Chemical shifts were calibrated on the D_2_O signal (*δ* = 4.21 ppm) [26]. A sample was filled into 5 mm NMR tubes and sodium 3-(trimethylsilyl)propane-1-sulfonate (DSS) was used as the internal standard to determine the integral intensity to calculate the conversion.

Solid-state NMR (ssNMR) spectra were collected using a 700 MHz Bruker Avance Neo NMR spectrometer (*B*_0_ = 16.4 T) at Larmor frequencies ν(^13^C) = 176.110 MHz using a double-resonance 3.2 mm magic angle spinning (MAS) probe. The 13C ssNMR experiments were performed at 20 kHz spinning speed with SPINAL 64 decoupling sequence. ^13^C CP/MAS NMR spectra were recorded with 1.5 ms spin-lock at 10,240 scans and 4 s recycle delay. The ^13^C MAS NMR experiments were performed using 3.45 µs at 154.3 W (π/2 pulse) with two distinct recycle delays 20 s (5120 scans) and 1 s (10,240 scans), respectively. The NMR experiment with a long relaxation delay was used for acquiring quantitative ^13^C MAS NMR spectra, whereas the NMR experiment with a short relaxation delay was used for selective detection of flexible parts of copolymer(s). The 13C chemical shifts were calibrated using α-glycine (176.03 ppm; carbonyl signal) as external standards. The sample was kept and packed into ZrO_2_ rotors under laboratory atmosphere and the Bruker TopSpin 3.2 pl7 software package (3 April 2017; https://www.bruker.com/protected/en/services/software-downloads/nmr/pc/pc-topspin.html) was used to process the spectra.

## 3. Results and Discussion

In general, nanogels are three-dimensional networks formed by cross-linked polymer chains prepared from hydrogel particles of nanometer sizes; thus, hydrogel and nanoparticle properties occur simultaneously in nanogels. Importantly, biodegradable nanogel-based nanomedicines should fulfill all the requirements for drug delivery systems. Herein, we present the tailored synthesis of biodegradable nanogels suitable as carriers of low-molecular-weight drug molecules.

### 3.1. Preparation of Nanogels—Effect of the BAC Crosslinking Comonomer Concentration

The reproducible dispersion polymerization conditions in the H_2_O/MetCel (80/20 *w*/*w*) mixture were evaluated to prepare biodegradable PHPMA-(Boc)BMH-BAC nanogels according to a slightly modified procedure, shown in Figure 1 [19]. In this study, the effect of biodegradable BAC crosslinking comonomer concentration varying from 5 to 25 wt% on the properties of PHPMA-BocBMH-BAC nanogels was investigated (Table 1). During all these experiments, the (Boc)-BMH concentration was constant (10 wt%) and the BAC concentration was increased from 5 to 25 wt% relative to the HPMA concentration that was decreased from 85 to 65 wt% (Table 1). According to TEM analysis, the PHPMA-BocBMH-5wt%BAC nanogel (NG1) contained irregular and broadly distributed tiny nanoparticles with *D*_n_ = 28 nm, *D*_w_ = 52 nm, and *Đ* = 1.83 (Figure 1a). The aqueous dispersion of NG1 measured with DLS showed a similar broad particle size distribution (PDI = 0.376) with Z-average *D*_h_ = 84.5 nm indicating the ability of NG1 to swell in aqueous medium (Table 1). An increase in BAC concentration to 10 wt% yielded a spherical NG2 nanogel with a narrower particle size distribution possessing *D*_n_ = 21 nm, *D*_w_ = 23 nm, and *Đ* = 1.14 (Figure 1b and Table 1). DLS measurement of the aqueous dispersion of the swollen NG2 nanogel showed that NG2 had a broad particle size distribution (PDI = 0.509) and its diameter was advantageously reduced to *D*_h_ = 41.8 nm with an increased BAC crosslinker concentration (Table 1). NG2 contained a small fraction of larger ~200 nm objects which scattered the light more than most smaller particles (Figure 1b), leading to a broadening particle size distribution [27].

We hypothesize that the presence of larger objects was a result of negligible aggregation during dispersion polymerization. A further increase in BAC concentration to 15 wt% resulted in a decreased diameter of NG3 with a spherical shape documented with *D*_n_ = 13 nm, *D*_w_ = 15 nm, and *Đ* = 1.15 (Figure 1c). The diameter decrease with the increase in BAC crosslinker concentration is attributed to the faster phase separation and the formation of smaller nanogels [28]. Surprisingly, swollen NG3 also showed an increased Z-average diameter (*D*_h_ = 79.1 nm). The increased swelling behavior of NG3 could be attributed to the decreased crosslinking density of the particle center and possible core–shell structure of NG3 [29]. The swollen NG3 also had a broad particle size distribution (PDI = 0.472) probably due to the presence of large objects as shown in Figure 1c and discussed for NG2. Similar spherical and uniform nanogels of NG4 with *D*_n_ = 22 nm, *D*_w_ = 25 nm, and *Đ* = 1.14 (Figure 1d) were obtained during copolymerization with 20 wt% BAC. However, NG4 contained a small fraction of aggregated nanogel. The aqueous dispersion of NG4 had a Z-average diameter *D*_h_ = 71.4 nm, documenting increased swelling behavior and a core–shell structure similar to NG3 [29]. Nevertheless, a more pronounced broadening of the particle size distribution of NG4 (PDI = 0.942) was observed, which was attributed to the presence of bulky aggregates (Figure 1d) which scattered more light compared to the major fraction of small nanogels in NG4 [27]. According to the TEM analysis, the increase in the BAC concentration to 25 wt% (NG5) led to the formation of the most aggregated ~70 nm objects which were composed of small ~17 nm nanogels. This observation was also confirmed with DLS analysis, documenting the presence of ~230 nm aggregated objects.

The A4F analyses of the swollen NG1-NG5 nanogels revealed the determined diameters of gyration (*D*_g_) and shape factors (*ρ* = *D*_g_/*D*_h_) (Table 1), showing that the *D*_g_ values of swollen NG1-NG5 were smaller in comparison to *D*_h_ from DLS and were in the range of 15.8 to 37.8 nm. The calculated *ρ* values ranged from 0.31 to 0.48, indicating that the PHPMA-BocBMH-BAC nanogels were comprised of a more crosslinked compact core with a less dense shell with heterogeneous polymer density [30,31,32]. Significant differences between *D*_h_ values (reflecting the outer dimension of the geometry) and *D*_g_ values (reflecting the localizations of mass with respect to the center of gravity of the molecule) caused by the densely crosslinked core of the PHPMA-BocBMH-BAC nanogels indicate their ability to swell in aqueous media [29,33].

### 3.2. Preparation of Nanogels—Effect of the Polymerization Medium Composition

To study the effect of polymerization medium composition on the size and morphology of PHPMA-BocBMH-BAC nanogels, we first varied the H_2_O/MetCel ratio (75/25 and 85/15 *w*/*w*) and secondly, we applied another solvent from the cellosolve family, 2-ethoxyethanol, in various ratios with H_2_O. For this purpose, we selected the conditions of dispersion polymerization of NG4 that were most appropriate for the preparation of colloidal stable core–shell PHPMA-BocBMH-BAC nanogels. Unfortunately, varying the H_2_O/MetCel ratio had a deteriorating effect on the PHPMA-BMH-BAC nanogels NG6 and NG7 (Figure 2a,b), with both nanogels containing large aggregates and a low fraction of individual nanogels.

The size and morphology of the PHPMA-BocBMH-BAC nanogels (NG8-NG10) prepared using various H_2_O/EtCel ratios from 75/25 to 85/15 are shown in Table 1. The dispersion polymerization in medium with H_2_O/EtCel ratios of 75/25 *w*/*w* and 80/20 *w*/*w* yielded slightly conjugated nanogels NG8 and NG9 (Figure 3a,b) with *D*_n_ = 20 nm and *D*_n_ = 23 nm, respectively. The highest EtCel content in the mixture with H_2_O (NG8; H_2_O/EtCel ratio = 75/25 *w*/*w*) formed PHPMA-BocBMH-BAC nanogels with a broad particle size distribution (*Đ* = 1.85) due to the higher solvency of the polymerization medium for the resulting polymer. With a decrease in EtCel content (H_2_O/EtCel ratio = 80/20 *w*/*w*), the particle size distribution of the NG9 nanogels was narrowed (*Đ* = 1.30), suppressing the formation of undesired larger aggregated particles (Table 1) [34]. DLS measurements showed that NG8 (*D*_h_ = 35.7 nm) and NG9 (*D*_h_ = 34.6 nm) were larger in comparison with TEM analyses, documenting their swelling ability in aqueous medium. A colloidally stable NG10 nanogel with an average diameter *D*_n_ = 36 nm and similar particle size distribution (*Đ* = 1.37) as NG9 was obtained with a further decrease in the EtCel content (H_2_O/EtCel ratio = 85/15 w/w) (Table 1). Importantly, no coagulum was formed in that case (Figure 3c). However, NG10 contained two families of particles with diameters *D*_n_ = 48 nm and *D*_n_ = 25 nm, respectively. The particle diameters of nanogels prepared in the H_2_O/EtCel polymerization mixture (NG8–10) decreased with increasing H_2_O due to a slower precipitation rate and reduced number of nuclei [33]. The DLS analysis revealed that NG10 had a similar hydrodynamic diameter (*D*_h_ = 34 nm) to NG8 and NG9 prepared in H_2_O/EtCel polymerization mixtures with a broad particle size distribution (Table 1).

The DLS and A4F analyses (Table 1) indicated that dispersion polymerization with various H_2_O/EtCel ratios produced core–shell type nanogels, as documented with *ρ* values 0.62 and 0.53 for NG8 and NG9, respectively. NG10 was closer to the hard particle with a constant internal polymer density (*ρ* = 0.71) [35], with little difference between the NG10 diameter in the dry state (*D*_n_ = 36 nm by TEM) versus the swollen state (*D*_h_ = 34 nm by DLS), indicating a homogeneous distribution of monomers in NG10.

In summary, colloidally stable and homogenous PHPMA-BocBMH-BAC nanogels with a regular shape were prepared in H_2_O/MetCel (80/20 *w*/*w*) and 20 wt% BAC as the crosslinking monomer (NG4). All the prepared nanogels exhibited a broad particle size distribution mainly due to the high concentration of the crosslinking BAC monomer and also due to the copolymerization of the three monomers at relatively high concentrations. This is in an agreement with our previous study regarding the fluorescently labeled PHPMA-EDMA-PMA nanogel for live cell imaging [19]. Nevertheless, the deprotection of the Boc/protected hydrazide groups in the nanogels became highly problematic during subsequent modification to obtain a colloidally stable nanogel. Thus, dispersion polymerization was employed to manufacture a reactive PHPMA-BMH-BAC nanogel containing free reactive hydrazide groups. The BMH monomer with deprotected hydrazides was employed using the optimized conditions: 70 wt% HPMA, 20 wt% BAC, and 10 wt% BMH in H_2_O/MetCel (80/20 *w*/*w*) stabilized with PVA and PVP and initiated with KPS under the same conditions as for the preparation of NG4 (Table 1). The NG11 nanogel was spherical with *D*_n_ = 30 nm, *D*_w_ = 35 nm, and *Đ* = 1.16 (Figure 4).

Zhou et al. also prepared biodegradable PHPMA micelles with a diameter ranging from 10 to 20 nm by self-assembly and cross-linked through the hydrazone linkages which had a similar shape and morphology [36]. DLS analysis of swollen NG11 decreased the size (*D*_h_ = 38 nm) compared to NG4 with BocBMH (*D*_h_ = 71.4 nm) and narrowed PDI = 0.494. For the comparison, we also evaluated NG11’s diameter using DLS under physiologic conditions (PBS buffer, pH 7.4; 37 °C) due to the intended application of the nanogels as drug delivery vectors, revealing that NG11 exhibited an increased *D*_h_ = 52 nm with surface zeta potential *ζ* = –11 mV probably due to a more expanded structure of the NG11 network at a higher temperature. A4F analysis determined *D*_g_ = 16.6 nm and *ρ* = 0.44 (Table 1), confirming that NG11 also had a crosslinked compact core with a core–shell structure [32]. Figure 2 illustrates the formation of the core–shell PHPMA-BMH-BAC nanogel by dispersion polymerization. The versatility of this dispersion polymerization was demonstrated for the preparation of hydrophilic nanogels of suitable sizes for application as carriers of low-molecular-weight bioactive compounds. Importantly, particles were prepared with smaller diameters than those described in the literature using optimized conventional dispersion polymerization [19,37].

### 3.3. Kinetic Study of Dispersion Polymerization

A ^1^H NMR kinetic study of nanogel NG11 was conducted to further understand the incorporation of each comonomer into the nanogels (Figure 5). The conversion of monomers was monitored over time for NG11 (70 wt% HPMA, 20 wt% BAC, and 10 wt% BMH in polymerization feed). A Steep initial polymerization rate up to 5 h was observed when the dispersion polymerization was almost complete, with the polymerization finishing after 15 h. The BAC and BMH monomer conversions reached 92% and 89%, respectively, thus showing high incorporation of those monomers and their higher copolymerization parameters. However, the conversion of HPMA was only 71% (Figure 5a). Similarly, Duracher et al. reported that precipitation polymerization of *N*-isopropyl methacrylamide and *N*,*N*′-methylenebisacrylamide (MBA) initiated with KPS was quicker and almost completed after 3 h due to the higher initiator concentration and copolymerization with a more reactive crosslinking monomer [38]. Similar polymerization kinetics were also observed during the preparation of poly(*N*-ethyl methacrylamide) microgels crosslinked with MBA or EDMA by emulsion/precipitation [39]. Figure 5b shows the experimental changes in the NG11 copolymer composition over time with the amount of monomer in the polymerization feed. As mentioned above, all three monomers were almost polymerized and incorporated into NG11 during the first 5 h. When the dispersion polymerization was complete, 50 wt% of HPMA, 19 wt% of BAC, and 9 wt% of BMH were incorporated into NG11. Almost all BAC and BMH monomers were incorporated into NG11 but 22 wt% HPMA was not copolymerized or only short soluble PHPMA oligomers were prepared and then removed during purification of NG11. This indicates that BAC and BMH monomers were more reactive and consumed much faster compared to HPMA. We hypothesize that this finding may reflect the calculated shape factor (*ρ* = 0.44) and crosslinked compact core with a less dense NG11 shell. Thus, the NG11 core is mainly composed of BAC crosslinking monomers with the main proportion of HPMA in the outer less-crosslinked, dense shell [40]. Indeed, such composition of the nanogels could be advantageous from the drug delivery point of view as the outer less-crosslinked hydrophilic shell could be highly biocompatible and thus guarantee the “stealth” properties of those nanogels during distribution to target tissue.

The ^13^C ssNMR spectroscopy (CP/MAS NMR and MAS NMR) was used to investigate the chemical composition and distribution of individual copolymers (morphology) in NG11 and NG4. Firstly, the absence of any signal(s) in the region between 110 and 150 ppm in the ^13^C MAS NMR spectra (Figure 6) indicates fully reacted/consumed double bonds and the formation of a three-dimensional polymeric network. Secondly, the absence of a narrow signal at ca. 25 ppm confirms the stability of the disulfide bridges. When the disulfide bridges are unstable and the thiol groups (-CH_2_-SH) are formed, the narrow signal at 25 ppm is present in the corresponding ^13^C MAS NMR spectrum (Figure 6).

The morphology of the nanoparticles was investigated by comparison of the ^13^C CP/MAS NMR spectra with the ^13^C MAS NMR spectra recorded with a short relaxation delay (1 s) (Figure 6). In principle, the ^13^C CP/MAS NMR experiment is designed to record the rigid parts of the samples, whereas ^13^C MAS NMR experiments with a short relaxation delay are used for the selective detection of flexible blocks of copolymer(s). From this comparison, it is evident that the most flexible part of the nanoparticles is a block with >CH-OH group from HPMA, and the corresponding peak is located ca 65 ppm. Moreover, in the NG4 system, the enhanced peak intensity around 20 ppm in the ^13^C MAS NMR spectrum is attributed to methyl groups (-CH_3_) of HPMA, indicating higher mobility of methyl groups in comparison to the NG11 system. Unfortunately, it is almost impossible to distinguish between BMH/BAC and hydrazine copolymers, indicating that the copolymers are relatively immobilized and most probably located in the nanoparticle core, confirming our hypothesis about the crosslinked compact core with a fuzzy shell.

### 3.4. Study of Nanogel Degradation

Various stimuli can be used for the degradation of drug delivery carriers, thus enabling their removal from the body after fulfilling their role of carrier. Tumor cells have a two orders of magnitude higher glutathione concentration compared to the blood [41]. Moreover, thioredoxin 1 is overexpressed in those tumor cells; thus, reductive degradation could be considered an important and suitable stimulus for the degradation. As NG11 contains a BAC crosslinker with disulfide linkage, we assessed NG11 nanogel degradation in the presence of 10 mM GSH in PBS buffer (pH 7.4) at 37 °C via TEM. The TEM image of the original NG11 nanogel showed a compact structure without significant structure disorder (Figure 7a). Importantly, a slightly distorted structure of NG11 was observed after 1 h exposure to GSH (Figure 7b). After 3 and 5 h, it was obvious that GSH continued to degrade the NG11 network; as the diameter decreased to *D*_n_ ~ 15 nm, the loss of compact NG11 nanogels was evident and the content of the decomposed structures increased (Figure 7c,d). After 24 h, NG11 appeared to be completely degraded since no individual NG11 nanogels were observed, only a continuous layer of decomposed NG11 (Figure 7e). These results indicate that NG11 was simultaneously and rapidly degraded by 10 mM GSH under physiologic conditions.

The reductive degradation of the nanogels together with their size of around 50 nm are the prerequisites for polymer-based systems for controlled drug delivery. In addition, suitable functional groups for connecting biologically active molecules were incorporated into the nanogel structure, so these systems should be tested for drug delivery.

## 4. Conclusions

We report the facile and reproducible procedure for the preparation of biodegradable and functional PHPMA-BMH-BAC nanogels with disulfide bonds by advanced dispersion polymerization. The optimal dispersion polymerization conditions for the preparation of hydrophilic nanogels are a mixture of H_2_O/MetCel (80/20 *w*/*w*) and the presence of a 20 wt% crosslinker. Importantly, the final PHPMA-BMH-BAC nanogel was even smaller than is typical for dispersion polymerization and compared to previously published results. The PHPMA-BMH-BAC nanogels had a diameter *D*_n_ = 30 nm in the dry state and swelling ability with a covalently crosslinked core–shell structure documented with DLS (*D*_h_ = 38 nm) and A4F analyses (*D*_g_ = 16.6). It was also completely degraded within 24 h by GSH under physiologic conditions studied with TEM. Thus, the developed nanogels have advantageous characteristics to become tailored carriers of various biologically active molecules and can be excreted from the body by glomerular filtration once they fulfill their role as drug carriers in tumorous cells.

## Data Availability

The data that support the findings of this study are available from the corresponding author upon reasonable request.

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
