# Peer review of "Biodegradable Covalently Crosslinked Poly[N-(2-Hydroxypropyl) Methacrylamide] Nanogels: Preparation and Physicochemical Properties"

_polymers, 2024, doi:10.3390/polym16020263_

Round 1

Reviewer 1 Report

Comments and Suggestions for Authors

The manuscript entitled " Biodegradable Covalently Crosslinked Poly[N-(2-Hydroxypro- 2 pyl) Methacrylamide] Nanogels: Preparation and Physicochemical Properties" is interesting research. The research reports the synthesis of PHPMA-BAC-BMH nanogel suitable for drug delivery study by studying the effect of BAC concentration, different ratios of H2O/MetCel and H2O/EtCel on the size, particle size distribution, morphology etc. The author provide a sufficient introduction section and the results of the research. The author clearly stated the definition with examples, mentioned the aim of the study and clearly explained the results and discussion. However, it needs minor revision before acceptance.

1.     How are the ratios of H2O/MetCel and H2O/EtCel were chosen?

2.     For the TEM images, please provide the size magnification.

3.     Figure 7c and 7d are unclear. Provide more clear images.

4.     For the synthesis of PHPMA-BAC-BMH, my suggestion is to add the spectroscopy analysis such as FTIR, GCMS, H-NMR if possible. This is for the confirmation of the polymer structure.

5.     Provide the schematic diagram/mechanism for the formation of PHPMA-BAC-BMH nanogel for better overview.

Author Response

Dear Reviewer,

Thank you very much for all the valuable comments and below we respond to them. We yellow-coloured text to highlight all changes in text.

  1. How are the ratios of H2O/MetCel and H2O/EtCel were chosen?

We selected the ratios according to our previous experience and work which we referred to in the submitted paper.

  1. For the TEM images, please provide the size magnification.

Please, TEM images have a scale bar which indicates the size of the nanoparticles. The value of the magnification is relative and can differ for different microscopes.

  1. Figure 7c and 7d are unclear. Provide more clear images.

Please, TEM images 7c and 7d show the polymer nanoparticles after partial degradation. Therefore, the degraded polymer spread over the entire carbon film of the microscopic grid, preventing the negative staining with uranyl acetate.

  1. For the synthesis of PHPMA-BAC-BMH, my suggestion is to add the spectroscopy analysis such as FTIR, GCMS, H-NMR if possible. This is for the confirmation of the polymer structure.

Please, in our study we did two independent analyses (1H NMR and solid-state NMR) two surely confirm the structure. The results and confirmation has been in our submitted paper.

  1. Provide the schematic diagram/mechanism for the formation of PHPMA-BAC-BMH nanogel for better overview.

Please, we added the schematic mechanism into the paper.

Reviewer 2 Report

Comments and Suggestions for Authors

This is an interesting study about biodegradable covalently crosslinked poly[N-(2-hydroxypropyl) methacrylamide] nanogels. I suggest it for publication after the following minor points are well addressed.

1. Line 28-29, Nanogels represent three-dimensional nanoscale structures with physically and/or covalently crosslinked polymer networks. One recent review (Gels 2022, 8(1), 46) should be included to support such a claim. 

2. For readers, one figure should be added about the nanogels including the chemical structures.

3. The PDIs measured by DLS in this study were pretty high (>0.3). The authors should add more discussion about this point.

4. Technical issues. For example, 'Dh, Dn, Dg' should be written as 'Dh, Dn, Dg'. Please check all.

Comments on the Quality of English Language

Minor editing of English language required

Author Response

Dear Reviewer,

Thank you very much for all the valuable comments and below we respond to them. We yellow-coloured text to highlight all changes in text.

  1. Line 28-29, Nanogels represent three-dimensional nanoscale structures with physically and/or covalently crosslinked polymer networks. One recent review (Gels 2022, 8(1), 46) should be included to support such a claim. 

Please, we completed our paper with this reference.

  1. For readers, one figure should be added about the nanogels including the chemical structures.

Please, we added the schematic mechanism into the paper.

  1. The PDIs measured by DLS in this study were pretty high (>0.3). The authors should add more discussion about this point.

Please, we added more discussion. But in general, it is the result of high concentration of crosslinking monomer and the copolymerization of three monomers. It is generally know that causes distortion of particle size distribution. And we also explained and discussed in our paper.

  1. Technical issues. For example, 'Dh, Dn, Dg' should be written as 'Dh, Dn, Dg'. Please check all.

Please, we corrected these mistakes.

Please, our article had been proofread in English.